# Factors Affecting Implant Salvage in Patients with Complications After Post-Mastectomy Implant-Based Reconstruction

**DOI:** 10.3390/jcm14082682

**Published:** 2025-04-14

**Authors:** Kyuseok Im, Siu-Yuan Huang, Yilan Jiangliu, Steven Yoshinaga, Albert Bai, Michael W. Chu, Antoine L. Carre, Anna M. Leung

**Affiliations:** 1Department of General Surgery, Kaiser Permanente Los Angeles Medical Center, 4867 W Sunset Blvd, Los Angeles, CA 90027, USA; kyuseok.x.im@kp.org (K.I.); yilan.x.jiangliu@kp.org (Y.J.); steven.x.yoshinaga@kp.org (S.Y.); 2Department of Breast Surgery, University of California, 10833 Le Conte Avenue, CHS 77-123, Los Angeles, CA 90095, USA; siuyuanhuang@gmail.com; 3Department of Research Support, Kaiser Permanente, 3280 E Foothill Blvd, Pasadena, CA 91107, USA; albert.x.bai@kp.org; 4Department of Plastic and Reconstructive Surgery, Southern California Permanente Medical Group, 6041 Cadillac Ave, Los Angeles, CA 90034, USA; michael.w.chu@kp.org; 5Department of Plastic and Reconstructive Surgery, City of Hope, 1500 East Duarte Road, Duarte, CA 91010, USA; acarre@coh.org; 6Department of Clinical Medicine, Pasadena Campus, Kaiser Permanente School of Medicine, 98 S Los Robles Ave, Pasadena, CA 91101, USA

**Keywords:** breast, breast cancer, breast surgery, mastectomy, breast reconstruction, breast implant, implant infection, reconstruction salvage, implant salvage, antibiotic treatment

## Abstract

**Background**: Implant-based reconstruction (IBR) is the most common method of breast reconstruction, but complications lead to patient distress and delays in cancer treatment. Management of implant complications is varied with no defined guidelines. The inability to salvage IBR is associated with infection, but the impact of antibiotics remains controversial. We aimed to analyze factors affecting salvage rates of threatened IBR requiring operative intervention. The primary outcomes were the rates of unplanned re-operation for threatened IBR for wound closure, exchange, or explant. We hypothesized antibiotic administration would improve salvage rates. **Methods**: A retrospective review of patients undergoing mastectomy with IBR from 2012 to 2023 was performed. Threatened IBR was defined as implant exposure, infection, skin necrosis, hematoma, seroma, or wound dehiscence without implant exposure. Management options for patients with implant infection included implant removal and antibiotic treatment, antibiotic treatment alone, implant replacement, washout and implant replacement, or implant removal without a salvage attempt. **Results**: In total, 6901 patients underwent post-mastectomy IBR, and 184 (2.7%) patients had an unplanned re-operation. A total of 166/184 patients (90.2%) underwent explantation, and 18/184 (9.8%) patients had implant salvage. Between the explant and salvage groups, there were no differences in patient demographics, oncologic treatments, or operative characteristics. The explant group had a higher rate of infection (77.7% vs. 22.2%, *p* < 0.0001). There was no difference in culture positivity or antibiotic administration history. **Conclusions**: Implant salvage is feasible but limited by infection. Antibiotic administration does not improve salvage rates. Patient factors, oncologic treatment factors, or operative factors do not impact the ability to salvage.

## 1. Introduction

Implant-based reconstruction (IBR) is the most common method of breast reconstruction after mastectomy, accounting for approximately 80 percent of breast reconstructions done in the United States [1]. Patients who receive implants after mastectomy for cancer or cancer prophylaxis have significantly more complications than those who have implants for cosmetic reasons [2].

Reconstruction offers significant psychological benefits to patients who have undergone mastectomy, and prior studies have demonstrated the impact of reconstruction on patients’ quality of life [3]. However, despite improvements in surgical technique and breast implant materials, reconstruction is not without complications and challenges [4]. Many of the cited psychological benefits of breast reconstruction are negated when complications occur [5]. These complications can lead to significant patient stress and can delay curative oncologic treatment.

IBR complications can lead to readmissions, reoperations, and the potential to compromise the final reconstructive outcome. Historically, complications after IBR, especially when associated with infection, have mandated implant explantation.

The most common IBR complications include poor wound healing, seroma, infection, skin necrosis, implant exposure, and hematoma [4,6,7,8]. The current management of IBR complications is varied, with no defined guidelines or validated clinical protocols [9]. Aggressive intervention can make IBR salvage possible and is reasonable in carefully selected patients [10]. The methods and principles of IBR salvage vary widely. An inability to salvage IBR is associated with infection, especially with atypical organisms, history of radiotherapy, and implant biofilm formation [11,12,13]. In the setting of infection, the role and impact of antibiotics on IBR salvage rates remain controversial [14].

The aim of this retrospective study is to analyze the factors affecting the salvage rates of threatened IBR requiring operative intervention. The primary outcomes were the rates of unplanned re-operation in threatened IBR for wound closure, exchange, or explant. The main variables of interest were the rates of IBR infection and antibiotic administration for salvage attempts prior to operative intervention. Secondary variables included patient characteristics, operative characteristics at index reconstruction, preoperative chemotherapy or radiation, and type of implant. We hypothesized that antibiotic administration would improve IBR salvage rates.

## 2. Materials and Methods

An IRB-approved, retrospective review of all patients undergoing mastectomy either for cancer or for cancer prophylaxis with IBR from 1 January 2012 to 1 January 2023 in a regional hospital system was performed. The study was conducted using Kaiser Permanente Southern California data from 16 hospitals. Patients were identified using CPT/ICD10 codes. Abstracted data included patient characteristics (age, BMI, self-reported race, comorbidities), oncologic treatment history (chemotherapy, radiation), mastectomy characteristics (laterality, nipple-sparing, surgeon specialty, incision type), implant characteristics (expander, expander volume, direct-to-implant, implant position, implant volume), presence of infection, culture data, antibiotic administration history (pre-salvage, post-salvage, duration, administration route), and IBR outcome (salvage, exchange, explant). Implant infection was defined by the following: (1) chart documentation by surgeon, (2) culture data, or (3) diagnosis codes. Threatened IBR was defined as implant exposure, infection, skin necrosis, hematoma, seroma, or wound dehiscence without implant exposure. Patients were censored at a post-operative follow-up of at least 12 months in order to include late-onset infections, defined as any infection occurring later than 6 months after index reconstruction. The type of expander or prosthesis was at the discretion of the plastic surgeon. The expander or implant was placed either in a pre-pectoral position or a submuscular pocket after mastectomy. Antibiotic prophylaxis using cefazolin (14–20mg/kg) was always administered prior to the index mastectomy and reconstruction. In patients allergic to beta-lactams, either vancomycin or clindamycin was administered. Management options for patients with implant infection were divided into the following: (1) implant explant and antibiotic treatment, (2) antibiotic treatment alone, (3) implant exchange, (4) washout and implant exchange, or (5) implant explant without a salvage attempt. All explants were included, for any cause, including infection, implant rupture, or exposure. There was no standardized regional protocol across surgeons and hospitals guiding explantation or salvage. Antibiotic regimens were based on surgeons’ clinical judgment and personal clinical preferences. Decisions to explant were based on surgeons’ clinical judgment and guided by salvage failure on antibiotics alone, systemic infectious symptoms requiring operative source control, frequency of prior operative salvage attempts, intraoperative evaluation of infection severity, or patient desire for explantation.

To evaluate the differences between groups, statistical analysis was performed employing mean, standard deviation, median, Kruskal–Wallis tests, and chi-square tests using 2023 Microsoft Excel and IBM SPSS Statistics software version 29.0.

## 3. Results

### 3.1. Patient Characteristics and Ability to Salvage

A total of 6901 patients underwent post-mastectomy implant-based reconstruction, and 184 (2.7%) patients had unplanned re-operations related to IBR. In total, 166/184 patients (90.2%) ultimately underwent explantation, and 18/184 (9.8%) patients underwent implant salvage. The average time from index reconstruction to explant was 12.2 months, including late infections (more than 6 months after index reconstruction). Between the explant group and salvage group, there were no significant differences in patient demographics or characteristics, including age at the index operation, age at the operation for threatened IBR, time between the index reconstruction and operation for threatened IBR, BMI, race, or comorbidities which included diabetes, HgbA1c, hypertension, nutritional status, and Charlson Comorbidity Index score (Table 1).

### 3.2. Oncologic Treatment Characteristics and Ability to Salvage

Between the explant group and salvage group, there was no difference in oncologic treatment history, including neoadjuvant chemotherapy rates and radiation rates prior to operation for threatened IBR. (Table 1).

### 3.3. Operative Characteristics and Ability to Salvage

Between the explant group and salvage group, there was no difference in operative characteristics, including laterality or type of index mastectomy (nipple-sparing vs. non-nipple-sparing). There was no difference in the surgical subspecialty performing the mastectomy (plastic/reconstructive surgery vs. general surgery). There was no difference in incision type (inframammary, lateral, or peri-areolar). There was no difference in the type of index reconstruction done (direct-to-implant, tissue expander, or flap surgery). There was no difference in implant position (sub-pectoral vs. pre-pectoral). There was no difference in implant volume, tissue expander fill volume, or proportion of tissue expander filled. (Table 2)

### 3.4. Antibiotic Usage, Administration, Duration, and Ability to Salvage

The explant group had a higher rate of infection (77.7% vs. 22.2%, *p* < 0.0001). However, there were no differences in culture positivity or antibiotic administration history (before or after operation for threatened IBR, duration, or administration route—enteral vs. intravenous) (Table 3).

Sensitivity and power analyses were not performed previously. On post hoc power analysis, the study was adequately powered for the incidence of infection (Table 4).

Among the pre-salvage/explant antibiotics regimens, the most common single-agent antibiotic regimens were cephalexin (41/184, 22.3%), trimethoprim–sulfamethoxazole (18/184, 9.8%), and clindamycin (8/184, 4.3%); 50/184 (27.2%) patients received combination regimens, and 42/184 (22.8%) did not receive any pre-salvage/explant antibiotics. Among the post-salvage/explant antibiotic regimens, the most common single-agent antibiotic regimens were cephalexin (26/184, 14.1%), trimethoprim–sulfamethoxazole (19/184, 10.3%), and ciprofloxacin (13/184, 7.1%).

There were 80 patients in total who had positive cultures taken at the time of surgery for salvage vs. explant. Of these, 72/80 (90%) patients had monomicrobial infections; the most common organisms were Staphylococcus aureus (19/80, 23.8%), Pseudomonas (18/80, 22.5%), and coagulase-negative staphylococcus (12/80, 15%). A total of 8/80 (10%) patients had polymicrobial infections. In 43/80 (53.8%) patients, the antibiotic regimen given prior to surgery to salvage vs. explant provided appropriate coverage for organisms appearing in the final operative cultures.

## 4. Discussion

The present work analyzes the factors affecting the ability to salvage implant-based reconstruction in a regional hospital-based system. In our study, we found that no patient factors, oncologic factors, mastectomy or reconstruction characteristics, or antibiotic usage patterns were predictive of the success of implant salvage. Proven infection was the main driver of IBR salvage failure requiring explantation. Rates of IBR complications requiring re-operation reported in the literature showed a wide range, from 4% within the first 30 days to 55% in the long term. In comparison, the rate of IBR complications requiring re-operation in this study (2.7%) was relatively low [15,16].

A 2021 meta-analysis by Kanapathy et al. of management of peri-prosthetic breast infection reported that of 1044 patients, 29% with mild infections were able to be treated with antibiotics, and 39% underwent surgical salvage, of which 84% were able to successfully retain the salvaged implant without infection recurrence; 35% of the patients underwent implant explantation; 39% underwent re-insertion of a new implant at a later date; and 4.99% experienced recurrence of infection requiring explantation [9]. Relative to the findings of Kanapathy et al., the salvage rate in an infection setting was very low. Among 133 patients who underwent re-operation for salvage/explant, only 4/133 (3.0%) patients were able to undergo salvage or exchange. However, these findings do not accurately capture the proportion of IBR infections that could be treated with antibiotics alone.

Contrary to expectation, there were no patient comorbidities, prior oncologic treatment modalities, or operative characteristics that significantly affected the ability for implant salvage.

Prior studies have demonstrated modality-dependent and dose-dependent adverse effects of radiation [17]. More specifically, within the realm of breast reconstruction, post-mastectomy radiation has been shown to be associated with poorer reconstruction outcomes, including reconstructive failure and infection [18,19]. In total, 61/184 (33.2%) patients underwent post-mastectomy radiation prior to salvage/explant operation. Patients in the salvage/exchange group had a higher rate of prior radiation (44.4% vs. 31.9%) compared to the explant group. While we may expect that a higher rate of prior radiation would translate to a higher rate of reconstructive failure, the history of prior radiation was not significantly different between groups. Future studies may investigate whether there is an association between radiation history and overall IBR complication rates, not only the ability to salvage.

Historically, comorbidities such as high BMI and diabetes have been associated with inability to salvage [20,21]. In addition, one would expect that prior history of neoadjuvant chemotherapy or radiation would lead to higher inabilities to salvage; however, that was not the case in this study.

This may suggest that the ability to salvage after IBR complications may be independent of patient-related or cancer treatment-related factors. Among operative factors, larger implant volume, nipple-sparing mastectomy, lower surgeon volume, and direct-to-implant reconstruction have been associated with IBR failure [22,23]. In this study, between the explant and salvage groups, there were no significant differences in the proportion of nipple-sparing mastectomies, regardless of whether the complication-side mastectomy was performed by a reconstructive surgeon, the incision type, implant volume, or implant position. Furthermore, the explant group had a higher proportion of patients who underwent pre-pectoral implant placement (26.5% vs. 5.9%). While this difference was not statistically significant, based on the findings of Scardina et al., we might have expected pre-pectoral implant placement to be protective [24]. However, given that this difference was not significant and that the explant group actually had higher rates of pre-pectoral implant placement, this may indicate that implant position does not significantly impact IBR complication rates or ability to salvage. This may indicate that the true drivers of IBR failure are independent of operative factors.

In this study, antibiotic regimen, administration, duration, and method of delivery were not associated with salvage rates. With infection-associated IBR complications, antibiotics were often empirically prescribed in attempts to avoid re-operation. While some studies indicate that antibiotics do not significantly improve salvage rates, other studies advocate tailoring antibiotics based on culture data [7,25]. Hence, the true impact of antibiotics on salvage rates remains unclear. In the majority of cases in the pre-salvage/explant setting, antibiotics were prescribed empirically, based on surgeons’ clinical judgment or personal clinical preference. In 15/184 (8.2%) patients, despite positive OR cultures, post-salvage/explant antibiotics were not given when operative source control was determined to be sufficient. Although antibiotics were not universally tailored based on cultures, clinical decisions were made by board-certified surgeons. Furthermore, with multiple surgeons across 16 different hospitals, this confounder may have been minimized. It may be helpful to develop regional antibiograms and antibiotic guidelines to guide clinical decisions in a more standardized fashion.

While this study did not investigate whether patients experienced any adverse effects from antibiotics, surgeons must be cognizant of these potential side effects. Empiric use of antibiotics without strong indications may incur unnecessary costs to patients and drive antibiotic resistance patterns [25,26,27]. While there is no single study comparing the impact of oral vs. intravenous antibiotics on breast reconstruction outcomes, various studies highlight the efficacy of intravenous antibiotics and the inadequate impact of oral antibiotics [25,28,29]. In this study, neither antibiotic timing (before vs. after operation for threatened IBR) nor route of administration (enteral vs. intravenous) was associated with successful salvage. This may indicate that antibiotics alone may not have a significant effect on implant salvage. Given this lack of association, surgeons should consider prescribing antibiotics based not primarily on suspicion of IBR infection but rather on objective indications such as clinical signs of systemic infection or positive cultures.

IBR complications may require re-operation for various reasons, ranging from skin necrosis to fulminant implant infection [30,31,32]. This study suggests that when IBR complications require operative management, explantation is often necessary, and rates of successful salvage remain low. Thus, surgeons can preemptively counsel patients that multiple operative salvage attempts may not be successful.

Implant explantation for reconstruction failure is the most conservative approach and has the benefit of avoiding multiple hospitalizations, decreasing the total number of procedures, and avoiding delays in oncologic treatment. However, this does not take into account the significant impact breast reconstruction failure has on patient body image, psychosocial well-being, and overall quality of life. Arguably, if patients agree, every effort should be made to pursue pathways that allow for reconstruction salvage.

The findings of this study are limited by its retrospective nature, particularly in evaluating the causality of factors leading to the inability to salvage IBR. Furthermore, as a retrospective study, this study is confounded by biases inherent to retrospective studies, such as selection bias and information bias. However, this was a data-driven study leveraging a large data set, comprising multiple institutions within the systems and multiple surgeons, which helped to minimize confounders.

We recognize the data capture for this study may not have fully captured office-based procedures for management of IBR complications, and diagnoses associated with these visits may have been misclassified as only “post-operative examination” rather than more specific diagnoses associated with IBR complications such as “infection”, “cellulitis”, or “implant exposure”. This study is also limited in that it may under-report the true incidences of unplanned operations for threatened IBR, as some implants are explanted or salvaged in the minor procedure room in clinics rather than the main operating room. Furthermore, this study did not capture the antibiotics patients may have received at other hospitals for IBR complications. However, Kaiser Permanente is a large, integrated network that includes urgent care clinics and emergency departments. The electronic medical record system connects patient data and charts across all types of encounters, providing excellent data capture and allowing for accurate evaluation of long-term outcomes. Furthermore, the electronic medical record system allows access to patient charts at partnered hospitals. These features of our integrated healthcare system and unified electronic medical record system may have helped to minimize confounders. The involvement of multiple surgeons across various surgical subspecialties (general surgery, surgical oncology, plastic/reconstructive surgery) may also limit confounders.

Selection bias was minimized in this study by employing the electronic medical system’s search function to extract all patients who had undergone IBR after mastectomy in the regional hospital system during the study period. Selection bias was further minimized by having the clearly defined outcome of whether IBR could be salvaged after complications. Information bias was minimized in this study by evaluating objective data and clinical outcomes, rather than using subjective data such as patient experiences as primary data sources. We recognize the risk of information bias remains in terms of the determination of “infection” being based on clinical evaluation and not always culture results; this was minimized by the fact that “infection” status was evaluated in different patients by multiple different surgeons across different surgical subspecialties based on reasonably reliable signs or symptoms such as documented fever, leukocytosis, cellulitis, or abscess. Otherwise, all other data metrics used in this study were objective measures clearly documented in the electronic medical record.

This study was also limited in its ability to evaluate the psychosocial impact of reconstruction failure. Future studies may include validated patient-reported outcomes measures such as BREAST-Q to better evaluate the psychosocial morbidity of IBR complications requiring re-operation. This may be done in a retrospective fashion, surveying patients who have undergone explantation without further reconstruction, or in a prospective fashion, surveying patients over time as surgeons encounter IBR complications.

## 5. Conclusions

Based on the present study, implant salvage can be performed, but it is highly limited by implant infection. No patient factors, treatment factors, or operative factors impact the ability to salvage. In addition, antibiotic administration is limited in its ability to impact salvage. In the future, we aim to develop an algorithm for the management of IBR complications to maximize rates of implant salvage.

## Figures and Tables

**Table 1 jcm-14-02682-t001:** Patient demographics, patient characteristics, and operative characteristics between explant and salvage/exchange groups.

	Explant N = 166	Salvage/Exchange N = 18	Total N = 184	*p*-Value
Age at index reconstruction				0.76 ^1^
Mean (SD)	55.1 (10.8)	55.4 (12.4)	55.1 (10.9)	
Median (Range)	54.5 (31–81)	60 (32–72)	55 (31–81)	
Age at operation for threatened IBR				0.76 ^1^
Mean (SD)	56.1 (10.5)	56.6 (12.2)	56.1 (10.7)	
Median (Range)	55 (31–81)	60.5 (34–72)	55 (31–81)	
Time between index reconstruction and operation for threatened IBR (months)				0.54 ^1^
Mean (SD)	12.2 (21.6)	13.0 (13.2)	12.3 (20.9)	
Median (Range)	4 (0–123)	12 (0–41)	4 (0–123)	
BMI				0.81 ^1^
Mean (SD)	29.1 (6.1)	28.6 (5.6)	29.1 (6.0)	
Median (Range)	28.2 (16.3–45.7)	30.3 (19.6–36.6)	28.4 (16.3–45.7)	
Self-reported race				0.11 ^2^
Asian/Pacific Islander	10 (6.0%)	0	10 (5.5%)	
Black	23 (13.9%)	6 (35.3%)	29 (15.8%)	
Hispanic	56 (33.7%)	5 (29.4%)	61 (33.3%)	
White	77 (46.4%)	6 (35.3%)	83 (45.4%)	
Not reported	0	1	1	
Alcohol usage				0.99 ^2^
Yes	85 (51.2%)	9 (52.9%)	94 (51.4%)	
No	70 (42.2%)	7 (41.2%)	77 (42.1%)	
Not reported	11 (6.6%)	1 (5.9%)	12 (6.6%)	
Diabetes	17 (10.2%)	3 (17.6%)	20 (10.9%)	0.35 ^2^
Hypertension	58 (34.9%)	6 (35.3%)	64 (35.0%)	0.98 ^2^
Charlson Comorbidity Index score				0.61 ^1^
Mean (SD)	1.6 (1.1)	1.4 (1.1)	1.6 (1.1)	
Median (Range)	1 (0–7)	1 (0–4)	1 (0–7)	
Charlson group				0.67 ^2^
Mild	140 (84.3%)	15 (88.2%)	155 (84.7%)	
Moderate	26 (15.7%)	2 (11.8%)	28 (15.3%)	
Albumin				0.45 ^1^
Mean (SD)	3.9 (0.5)	4.0 (0.3)	3.9 (0.4)	
Median (range)	3.9 (2.3–4.7)	4.0 (3.6–4.5)	3.9 (2.3–4.7)	
Absolute neutrophil count				0.12 ^1^
Mean (SD)	4.3 (2.1)	3.5 (1.8)	4.3 (2.1)	
Median (range)	3.9 (0.5–15.4)	3.0 (1.0–6.8)	3.8 (0.6–15.4)	
Blood urea nitrogen				0.51 ^1^
Mean (SD)	12. (5.1)	11.8 (3.8)	12.8 (5.0)	
Median (range)	12 (3–33)	12 (6–19)	12 (3–33)	
Creatinine				0.63 ^1^
Mean (SD)	0.8 (0.2)	0.8 (0.2)	0.8 (0.2)	
Median (range)	0.8 (0.5–2)	0.8 (0.4–1.3)	0.8 (0.4–2)	
HgbA1c				0.34 ^1^
Mean (SD)	5.7 (0.6)	5.8 (0.4)	5.8 (0.6)	
Median (range)	5.7 (4.2–8.7)	5.8 (5.3–6.5)	5.7 (4.2–8.7)	
Chemotherapy preceding IBR complication	74 (44.6%)	8 (44.4%)	82 (44.6%)	0.99 ^2^
Radiation preceding IBR complication	53 (31.9%)	8 (44.4%)	61 (33.2%)	0.28 ^2^

^1^ Kruskal–Wallis. ^2^ Chi-square.

**Table 2 jcm-14-02682-t002:** Operative characteristics between explant and salvage/exchange groups.

	Explant N = 166	Salvage/Exchange N = 18	Total N = 184	*p*-Value
Threatened IBR laterality				0.09 ^2^
Bilateral	23 (13.9%)	2 (11.1%)	25 (13.6%)	
Left	67 (40.4%)	12 (66.7%)	79 (42.9%)	
Right	76 (45.8%)	4 (22.2%)	80 (43.5%)	
Index reconstruction: Nipple-sparing mastectomy	32 (19.3%)	6 (33.3%)	38 (20.7%)	0.16 ^2^
Index reconstruction: Bilateral mastectomy	114 (68.7%)	11 (61.1%)	125 (67.9%)	0.51 ^2^
Surgical subspecialty performing the mastectomy on threatened IBR breast side				0.56 ^2^
Plastics	6 (3.6%)	0 (0.0%)	6 (3.3%)	
Breast	156 (94.0%)	18 (100.0%)	174 (94.6%)	
Combo	4 (2.4%)	0 (0.0%)	4 (2.2%)	
Incision type				0.45 ^2^
IMF	16 (9.6%)	3 (16.7%)	19 (10.3%)	
Lateral	14 (8.4%)	0	14 (7.6%)	
Peri-areolar	133 (80.1%)	15 (83.3%)	148 (80.4%)	
Vertical	3 (1.8%)	0	3 (1.6%)	
Reconstruction at time of mastectomy				0.74 ^2^
DTI	16 (9.6%)	3 (16.7%)	19 (10.3%)	
TE	136 (81.9%)	15 (83.3%)	151 (82.1%)	
Flap	2 (1.2%)	0	2 (1.1%)	
No reconstruction	12 (7.2%)	0	12 (6.5%)	
Implant position				0.06 ^2^
Sub-pectoral	122 (73.5%)	16 (94.1%)	138 (75.4%)	
Pre-pectoral	44 (26.5%)	1 (5.9%)	45 (24.6%)	
Implant volume				0.52 ^1^
Mean (SD)	595 (174)	559 (188)	588 (176)	
Median (range)	595 (300–960)	560 (320–800)	595 (300–960)	
TE fill volume				0.16 ^1^
Mean (SD)	229 (161)	155 (175)	225 (162)	
Median (range)	200 (0–750)	100 (0–480)	200 (0–750)	
Proportion of TE filled				0.07 ^1^
Mean (SD)	0.4 (0.3)	0.2 (0.2)	0.4 (0.3)	
Median (range)	0.4 (0–1)	0.2 (0–0.6)	0.4 (0–1)	

^1^ Kruskal–Wallis. ^2^ Chi-square.

**Table 3 jcm-14-02682-t003:** Infection and antibiotic administration differences between explant and salvage/exchange groups.

	Explant N = 166	Salvage/Exchange N = 18	Total N = 184	*p*-Value
Infection	129 (77.7%)	4 (22.2%)	133 (72.3%)	<0.0001 ^2^
Culture taken during operation for threatened IBR—Positive	77 (55.4%)	3 (37.5%)	80 (54.4%)	0.32 ^2^
Antibiotics given before operation for threatened IBR	131 (78.9%)	11 (61.1%)	142 (77.2%)	0.09 ^2^
Route of administration				0.94 ^2^
Enteral	27/131 (20.6%)	2/11 (18.2%)	29 (20.4%)	
Intravenous	96/131 (73.3%)	9/11 (81.8%)	105 (73.9%)	
Both	8/131 (6.1%)	0	8/184 (4.3%)	
Duration (days)				0.25 ^1^
Mean (SD)	7.5 (7.5)	4.5 (3.2)	7.3 (7.3)	
Median (range)	7 (1–49)	4 (1–10)	6 (1–49)	
Antibiotics given after operation for threatened IBR	109 (65.7%)	9 (50.0%)	118 (64.1%)	0.19 ^2^
Route of administration				0.64 ^2^
Enteral	7/109 (6.4%)	0	7/118 (5.9%)	
Intravenous	99/109 (90.8%)	9/9 (100.0%)	108/118 (91.5%)	
Both	3/109 (2.8%)	0	3/118 (2.5%)	
Duration (days)				0.88 ^1^
Mean (SD)	11.9 (16.6)	10.0 (2.9)	11.7 (15.9)	
Median (range)	10 (4–180)	10 (5–14)	10 (4–180)	

^1^ Kruskal–Wallis. ^2^ Chi-square.

**Table 4 jcm-14-02682-t004:** Post hoc power analysis of the incidence of infection.

Power Analysis Components	Results
Incidence of infection—explant group	77.7% (129/166)
Incidence of infection—salvage/exchange group	22.2% (4/18)
Explant group size	166
Salvage/exchange group size	18
Alpha	0.05
Post hoc power	99.9%

## Data Availability

The raw data supporting the conclusions of this article will be made available by the authors on request.

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
