# Peer review of "Factors Affecting Implant Salvage in Patients with Complications After Post-Mastectomy Implant-Based Reconstruction"

_jcm, 2025, doi:10.3390/jcm14082682_

Round 1
Reviewer 1 Report
Comments and Suggestions for Authors
This study analyzed the factors affecting the salvage rate of implant-based breast reconstruction (IBR) and found that infection is the main factor limiting salvage, while antibiotic use has no significant impact on the salvage rate.
1.Sample Size Issue: The author mentions that the study is based on a regional hospital. Could you clarify the name of the hospital? How is it possible for a regional hospital to have 9,601 cases over 10 years? Please explain.
2. Insufficient detail on antibiotic use: The article mentions that the use of antibiotics does not significantly improve implant salvage rates, but does not provide detailed information on antibiotic regimens,,suggesting a more in-depth discussion of specific clinical approaches to antibiotic use.
3. Limitations of the study methodology: This is a retrospective study and there may be selection bias and information bias which may affect the accuracy of the findings. It is recommended that the authors describe this in detail in the discussion section.
4. Follow-up and Data Adequacy: The article mentions that the follow-up period was at least 12 months, but does not clarify whether late-onset infections were considered during this period. Additionally, radiation therapy is common after breast cancer surgery, and the relationship between radiation therapy and the study conducted by the authors should be discussed. Please reference the paper DOI: 10.1016/j.tranon.2024.102012 in this context.
5. Depth of Comparative Analysis: The authors mention several complications related to breast augmentation but lack bibliometric support for these claims.
Comments on the Quality of English Language/
Reviewer 2 Report
Comments and Suggestions for Authors
This retrospective analysis compares 184 threatened implant-based reconstructions (IBR) after mastectomy to identify the predictors of successful implant salvage. Reoperation and explantation rates were the primary outcomes of interest, and patient demographics, oncologic therapies, operative details, infection status, and antibiotic regimens were secondary variables. The overall finding is that infection is the greatest predictor of reconstructive failure, and no patient, oncologic, or operative factors—including antibiotic regimens—significantly influenced salvage success.
Strengths
Clinical Relevance – The paper addresses a crucial and understudied clinical situation in reconstructive breast surgery: salvageability of IBR amidst complications. This directly relates to surgical planning, patient counseling, and health system resource allocation.
Major comments
Low Salvage Rate and Small Salvage Cohort: Implant salvage was successful in only 18 of 184 patients (9.8%), limiting statistical power for subgroup comparisons. Include sensitivity an power analyses in the Methods section. If you do not have done it before, perform a post-hoc power analysis. This is mandatory.
Limited Reporting on Microbiological Data: While infection was the leading reason for explantation, no information is given in the manuscript on infecting organisms, resistance patterns, or rationale for antibiotic choice. Report on culture isolates and directed antibiotic therapy to more critically assess why antibiotic use did not impact outcomes.
Recent Literature: There is a lack of recent data and literature in the Discussion section. Consider expanding it with adding references such as PMID: 36143318 to improve your manuscript.
Unclear Criteria for Salvage Attempts: The manuscript does not explain how salvage versus explant decisions were made, or whether clinical protocols were standardized across surgeons and centers. Establish institutional or surgeon-specific criteria for pursuing salvage to facilitate interpretation of variability in outcomes.
Retrospective Design and Absence of Patient-Reported Outcome Data: The retrospective design limits causality, and the psychosocial impact of reconstruction failure—considered in the introduction—is not assessed. Future studies should include validated patient-reported outcome measures (PROMs) such as BREAST-Q and consider prospective designs.
Round 2
Reviewer 1 Report
Comments and Suggestions for Authors
I think the suthor has did a good job, congratulations!
Reviewer 2 Report
Comments and Suggestions for Authors
Thank you for your revision. No more comments.